# Preconception Folic Acid Supplement Use in Immigrant Women (1999–2016)

**DOI:** 10.3390/nu11102300

**Published:** 2019-09-27

**Authors:** Roy M. Nilsen, Anne K. Daltveit, Marjolein M. Iversen, Marit G. Sandberg, Erica Schytt, Rhonda Small, Ragnhild B. Strandberg, Eline S. Vik, Vigdis Aasheim

**Affiliations:** 1Faculty of Health and Social Sciences, Western Norway University of Applied Sciences, Inndalsveien, 28, 5063 Bergen, Norway; marjolein.memelink.iversen@hvl.no (M.M.I.); erica.schytt@ltdalarna.se (E.S.); ragnhild.bjarkoy.strandberg@hvl.no (R.B.S.); eline.skirnisdottir.vik@hvl.no (E.S.V.); vigdis.aasheim@hvl.no (V.A.); 2Department of Global Public Health and Primary Care, University of Bergen, Kalfarveien 31, 5018 Bergen, Norway; anne.daltveit@uib.no; 3Division of Health Data and Digitalisation, Norwegian Institute for Public Health, Zander Kaaesgate 7, 5018 Bergen, Norway; 4Department of Obstetrics and Gynecology, Haukeland University Hospital, Jonas Lies vei 72, 5053 Bergen, Norway; marit.g.sandberg@gmail.com; 5Centre for Clinical Research Dalarna, Uppsala University, Nissers väg 3, 791 82 Falun, Sweden; 6Department of Women’s and Children’s Health, Karolinska Institute, 171 77 Stockholm, Sweden; r.small@latrobe.edu.au

**Keywords:** country of birth, ethnicity, folate, folic acid, immigrant, length of residence, migrant, neural tube defects, Norway, pregnancy, vitamins

## Abstract

This study examines how preconception folic acid supplement use varied in immigrant women compared with non-immigrant women. We analyzed national population-based data from Norway from 1999–2016, including 1,055,886 pregnancies, of which 202,234 and 7,965 were to 1st and 2nd generation immigrant women, respectively. Folic acid supplement use was examined in relation to generational immigrant category, maternal country of birth, and length of residence. Folic acid supplement use was lower overall in 1st and 2nd generation immigrant women (21% and 26%, respectively) compared with Norwegian-born women (29%). The lowest use among 1st generation immigrant women was seen in those from Eritrea, Ethiopia, Morocco, and Somalia (around 10%). The highest use was seen in immigrant women from the United States, the Netherlands, Denmark, and Iceland (>30%). Folic acid supplement use increased with increasing length of residence in immigrant women from most countries, but the overall prevalence was lower compared with Norwegian-born women even after 20 years of residence (adjusted odds ratio: 0.63; 95% confidence interval: 0.60–0.67). This study suggests that immigrant women from a number of countries are less likely to use preconception folic acid supplements than non-immigrant women, even many years after settlement.

## 1. Introduction

Maternal use of folic acid supplements before and in early pregnancy can prevent a child from having a neural tube defect (NTD) [1]. This has led numerous countries worldwide to fortify their foods with folic acid to increase the uptake of this B vitamin in women of reproductive age [2]. In the European Union, there is currently no mandatory food fortification with folic acid [3], but health authorities in many European countries recommend that all women planning a pregnancy should begin folic acid supplementation before becoming pregnant, in addition to consuming food folate from a varied diet [3,4].

Despite these recommendations having been in place for years, the use of preconception folic acid supplements among pregnant women is overall still low, and many women start supplementation too late with respect to NTD prevention [5,6]. Among women with the lowest use are those with unintended pregnancies, young women, and those with the lowest levels of education [6,7,8]. In addition, folic acid supplement use varies considerably among countries [9], and several studies suggest that being an immigrant or having a foreign background is a strong determinant for low use [10,11,12,13,14,15,16]. The low use in immigrants is of particular concern, because women from some countries may have higher NTD occurrence than others [17].

Of the studies of folic acid supplement use in immigrant women, however, few have provided data on individual women’s countries of birth, which is the preferred indicator for monitoring perinatal health in immigrant women [18]. In addition, there is limited information concerning the relationship between length of residence and folic acid supplement use [12], and on how folic acid supplement use varies across generations of immigrant women [19]. Investigating the associations with length of residence and generational status is also of relevance because it may provide valuable insight into whether the uptake of folic acid supplements by immigrants is increasing over time.

Using a large, national, population-based dataset in Norway, we report on folic acid supplement use in immigrant women for the three migration indicators: generational immigrant category, maternal country of birth, and maternal length of residence. Using this as a basis, we highlight which specific groups are in greatest need of intervention programs to increase folic acid supplement use for NTD prevention.

## 2. Materials and Methods

### 2.1. Setting

The Norwegian recommendations on folic acid supplement use for the prevention of NTD were issued in 1998 [20]. The recommendations state that all women planning a pregnancy or who may become pregnant should take a daily folic acid supplement of 0.4 mg from one month before pregnancy and throughout the first two to three months of pregnancy. Women with a previous NTD-affected baby, or those who themselves or whose partner have an NTD, are advised to take higher folic acid doses. Although based on small numbers, a reduction in the number of NTDs has been reported since the folic acid recommendations were issued in Norway [21].

### 2.2. Study Population

The present study drew on resources from the Medical Birth Registry of Norway (MBRN) and Statistics Norway (SSB) for the period 1999–2016 [22,23]. The data were linked using the national identity number, which is given to all Norwegian citizens and to foreigners with a permit to stay in Norway. The study unit of interest was the women’s pregnancy. Hence, the use of folic acid supplements was counted only once in women with multiple births (the proportion of multiple births was 1.77%). Initially, our study comprised 1,056,396 non-terminated pregnancies after gestational week 12. Of these, we excluded 510 pregnancies where information on the country of birth for the mother was missing, leaving data on 1,055,886 pregnancies and 612,054 women for analysis.

### 2.3. Vitamin Supplement Use

Data on the maternal use of folic acid supplements were obtained from the MBRN. The data were collected from the women by the attending caregivers at birth or during pregnancy at antenatal health visits. The data collection included information on vitamin type (folic acid and/or multivitamins) and timing of use (before and/or during pregnancy), but not on dose or frequency of supplement use. From 1999 onwards, most folic acid supplements contained 0.4 mg folic acid, whereas most multivitamin supplements contained 0.2 mg of folic acid [21]. We defined folic acid use as any supplemental use of folic acid or multivitamin that had started before the pregnancy (i.e., preconception use). All other reports were coded as no use.

### 2.4. Immigrant Category

Maternal immigrant category was obtained from SSB and included information on both the mother’s and her parents’ birthplace to reflect whether she was a non-immigrant (Norwegian-born woman with two Norwegian-born parents), 1st generation immigrant (foreign-born woman with two foreign-born parents), or 2nd generation immigrant (Norwegian-born woman with 1st generation immigrant parents). While these groups were our primary focus for analysis, we also provide information on folic acid supplement use according to the three remaining large groups of women with mixed background (see Table 1 below).

### 2.5. Maternal Country of Birth

Maternal country of birth among 1st generation immigrant women was obtained from SSB and was analyzed directly as a categorical variable including the 40 largest maternal countries of birth, i.e., those contributing to the most births during the study period. All other countries were grouped and analyzed as other countries. Based on information on women’s country of birth, we also analyzed immigrant women according to the seven Global Burden of Disease super regions [24]: Central Europe, Eastern Europe, and Central Asia; High-income; Latin America and Caribbean; North Africa and Middle East; South Asia; Southeast Asia, East Asia, and Oceania; and Sub-Saharan Africa.

### 2.6. Length of Residence

Length of residence was defined as the year of childbirth in the MBRN minus the year of immigration to Norway. The year of immigration was extracted from the registration date when immigrants are allocated a national identity number at the National Registry [25]. This number is essential for all public correspondence in Norway, and is only provided to foreigners with a valid residence permit of more than six months [25]. Length of residence was analyzed as a categorical variable using the time categories chosen a priori: <1, 1–2, 3–4, 5–10, 11–20, >20 years.

### 2.7. Other Variables

Additional variables derived were year of birth, maternal age at the time of birth, parity (0, 1, 2, 3, ≥4 previous births), marital status at birth (married, partner, single/other), maternal income (quartiles calculated for the whole study period), and maternal educational level (no education, primary education (1–10 years), secondary education (11–14 years), university/college (14–20 years)). All these variables have been previously associated with supplemental folic acid use in Norway [8], and were included as adjustment variables in the present analyses. We also adjusted for Norwegian health region (south-east, west, middle, north) to account for potential geographical variation in supplemental folic acid use.

### 2.8. Statistical Analysis

To examine the associations of immigrant category, country of birth, and length of residence with folic acid supplement use, we estimated odds ratios (ORs) with 95% confidence intervals (CIs) using binary logistic regression analyses. We incorporated immigrant category, country of birth, and length of residence in the regression models as independent categorical variables using Norwegian-born women with two Norwegian-born parents (i.e., non-immigrants) as the reference category.

We calculated both crude and adjusted ORs with 95% CIs. The adjustment variables were year of birth, maternal age, marital status, parity, health region, maternal income, and education. To account for non-linear relations, we included the continuous variables year of birth and maternal age at birth as polynomial quadratic model terms. To account for dependence among pregnancies by the same mother, we used robust standard errors (i.e., clustered Sandwich estimator) that allowed for within-mother clustering.

We performed the analyses in R 3.4.2 for Windows [26]. To avoid list-wise deletion and potential bias due to missing data in covariates (missing data are provided in Table 1), we used the predictive mean matching algorithm to create five multiply imputed datasets based on the rms-package in R [27]. The imputation model included the same variables as those contained in the final analytical models. The pooling of ORs with 95% CIs across datasets were performed using Rubin’s combination rules.

### 2.9. Ethics Approval and Consent to Participate

The study has been approved by the South East Regional Committees for Medical and Health Research Ethics in Norway (Folder number: 2014/1278). The approval includes the use of individual record linked data from the Medical Birth Registry of Norway and Statistics Norway.

## 3. Results

### 3.1. Sample Characteristics

Of the 1,055,886 pregnancies included in our study from 1999 to 2016, 19.2% (*n* = 202,234) were to 1st generation immigrant women and 0.8% (*n* = 7,965) to 2nd generation immigrant women. Compared with non-immigrant women, 1st and 2nd generation immigrant women overall had a higher proportion of the lowest educational level (Table 1). First generation immigrant women overall also had the highest proportion of the lowest income category, while 2nd generation immigrant women had younger age at birth and were more often primiparous (Table 1).

### 3.2. Folic Acid Supplement Use by Immigrant Category

Folic acid supplement use overall was reported by 21% and 26% of 1st and 2nd generation immigrant women respectively (Table 1). In comparison, the prevalence of folic acid supplement use was 29% in non-immigrant women and 29–31% in those of mixed background. There was a marked increase in folic acid supplement use in all groups from 1999 to 2016 (Figure 1). However, the lower use among 1st and 2nd generation immigrant women compared with non-immigrant women remained constant throughout the study period, also after adjusting for covariates in logistic regression models (Appendix A). The reported folic acid supplement use for the most recent period (2014–2016) was as follows: Norwegian-born: 43%; 2nd generation immigrants: 38; 1st generation immigrants: 32%.

### 3.3. Distribution of 1st Generation Immigrant Women

The remaining results will focus on folic acid supplement use among 1st generation immigrant women who came from more than 200 countries during the study period 1999–2016. Notably, 1st generation immigrant women from high-income countries, and Central Europe, Eastern Europe, and Central Asia constituted the largest group overall (Table 2), and women from Central Europe, Eastern Europe, and Central Asia had the largest increase in births during the period (data not shown). However, the five largest specific country groups were women from Somalia (7.4%), Poland (7.1%), Sweden (6.1%), Iraq (5.3%), and Pakistan (5.0%).

### 3.4. Folic Acid Supplement Use by Country of Birth in 1st Generation Immigrant Women

The reported folic acid supplement use in 1st generation immigrant women was lowest among women from Sub-Saharan Africa and those coming from North Africa and the Middle East (Table 2). When investigating folic acid supplement use by specific countries, the prevalence varied from 8% in Somali women to 41% in women from the United States (Figure 2). The lowest use, in terms of adjusted ORs, was observed in immigrant women from Eritrea, Ethiopia, Morocco, and Somalia. The highest use was observed in immigrant women from the United States, the Netherlands, Denmark, and Iceland. These associations remained essentially the same when restricting the analyses to the three last study years 2014–2016 (Appendix A).

### 3.5. Folic Acid Supplement Use by Length of Residence in 1st Generation Immigrants

With the exception of a few groups with initially high use (Denmark, Great Britain, Iceland, Netherlands, and Poland), women from nearly all countries increased their use with length of residence in Norway (Figure 3). Still, in overall analyses, folic acid supplement use in immigrant women was lower than that of Norwegian-born women, even after 20 years of residence (Table 3; OR 0.63 (0.60–0.67)). The slowest increase in prevalence in groups with initially low use was seen in women from China, India, Morocco, Turkey, and Vietnam.

## 4. Discussion

Maternal use of folic acid supplements before and in early pregnancy is important for the prevention of NTD in offspring. We found that the prevalence of reported preconception folic acid supplement use was lower overall in 1st and 2nd generation immigrant women compared with non-immigrant women. The lower use overall among 1st generation immigrant women was observed in women from most studied countries. Importantly, folic acid supplement use increased with increasing length of residence, suggesting that many women adopt the official folic acid recommendations over time. However, this increase was slow, and the overall use after 20 years of residence had still not reached the same levels as that among non-immigrant women.

This was a national, population-based study comprising data from all women giving birth in Norway over an 18-year period. The long time period allowed trend analyses to be undertaken, as well as analyses of specific immigrant groups arriving at different times. Further, we performed comprehensive adjustment for covariates in regression analyses to justify comparisons between immigrant groups and non-immigrants. However, our results may have limitations. One concern is that many immigrant women do not speak English, and are thus unfamiliar with the English term “folic acid” or “folate”. In the absence of an interpreter during data collection, such language barriers may potentially have led to underreporting of folic acid supplement use in some immigrant women. Furthermore, folic acid supplement use in immigrant and non-immigrant women in Norway may not be representative of similar women in other countries. Therefore, the estimated prevalence and ORs of folic acid supplement use in the present study may not be entirely generalizable to other countries, overall, or for different time periods.

Our study provides more in-depth information regarding folic acid supplement use in immigrant women than previous studies [10,11,12,13,14,15,16,19]. Previous investigations have typically reported data according to large heterogeneous groups, potentially masking a great deal of variation in immigrant subgroups, and are thus unable to precisely identify where strategies to increase preconception use of folic acid supplements should be focused. In particular, we found that only around 10% of immigrant women from Eritrea, Ethiopia, Morocco, and Somalia had used folic acid supplements, whereas more than 30% of women from the United States, the Netherlands, Denmark, and Iceland had done so over the study period. We also detected considerable variation within continents; for example in Europe, women from the Netherlands had about four times higher prevalence than women from Kosovo, which further underscores the importance of our country-specific analyses.

It is conceivable that different use across immigrant women’s countries of birth may reflect different levels of folic acid knowledge in immigrant women prior to migrating to Norway. For example, the United States, Great Britain, and the Netherlands were among the first countries to issue recommendations on folic acid supplement use for the prevention of NTDs [2,4]. Already in the 1990s, the United States and other countries globally also introduced widespread mandatory food fortification with folic acid to increase blood folate concentration in women [2]. In these countries, a reduction of NTD incidence has been reported [2]. In contrast, many countries worldwide still do not have food fortification programs or recommendations for folic acid supplementation. Use of folic acid supplements in women coming from these countries thus relies on how well they receive or adopt folic acid recommendations from caregivers in post-migration countries.

To obtain some data on how well information on folic acid supplement use was received or adopted by immigrant women over time, we analyzed folic acid supplement use according to length of residence of immigrant women. With the exception of a few countries with initially high use, women from almost all countries increased their use over time. Although the increase was slow overall, this is a promising finding, because it shows that it might be feasible through effective campaigns and promotions to increase use during the earlier settlement period as well. Our results partly agree with a similar study from Canada showing a distinct increase in the use of folic acid supplements with length of residence [12]. However, that study was considerably smaller than ours, and it did not show how folic acid supplement use varied within specific maternal countries of birth.

Our results show that the prevalence of preconception folic acid supplement use has increased in both immigrant and non-immigrant women since the recommendations were issued in 1998, but that it remains low in all groups. National information campaigns may be a means for increasing preconception folic acid supplement use in the overall population, but may fail to reach the subgroups which are most in need of increased uptake [7]. In Norway, 29% (15,766/55,120) of live births in 2018 were to immigrant women [28], and this percentage is likely to increase in light of growing international migration. Consequently, in countries where mandatory food fortification is not yet permitted, strategies for improving recommended folic acid supplement use in vulnerable immigrant women are urgently needed and may have important NTD preventive effects.

We also examined how folic acid supplement use varied across generations of immigrant women. The finding of a consistently lower prevalence of folic acid supplement use in 2nd generation immigrant women was somewhat surprising, and is in contrast to the findings of a study from France [19]. According to previous studies, 2nd generation immigrants are often more similar to non-immigrants than to 1st generation immigrants in behavior and lifestyle [29,30]. On this basis, we had expected smaller or no difference in folic acid supplement use between 2nd generation immigrant and Norwegian-born women. Explanations for the lower use in 2nd generation immigrant women in Norway should be further explored in order to understand ways of increasing use among these women.

## 5. Conclusions

This national population-based study from Norway shows that immigrant women from a large number of countries use folic acid supplements to a lesser degree than non-immigrants. In addition, we found that this lower use remained, even many years after settlement. This suggests that the current recommendations and intervention programs to increase folic acid supplement use in fertile women should be better adapted for immigrant women in Norway.

## Figures and Tables

**Figure 1 nutrients-11-02300-f001:**
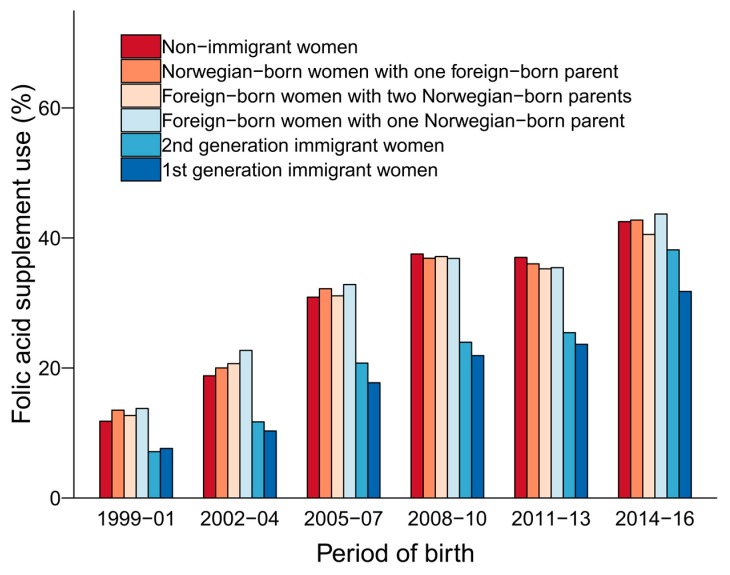
Preconception folic acid supplement use by maternal immigration category (1999–2016). Numbers, percentages and odds ratios for preconception folic acid supplement use are shown in Appendix A. Non-immigrant women: Norwegian-born women with two Norwegian-born parents; 2nd generation immigrant women: Norwegian-born women with two foreign-born parents; 1st generation immigrant women: Foreign-born women with two foreign-born parents.

**Figure 2 nutrients-11-02300-f002:**
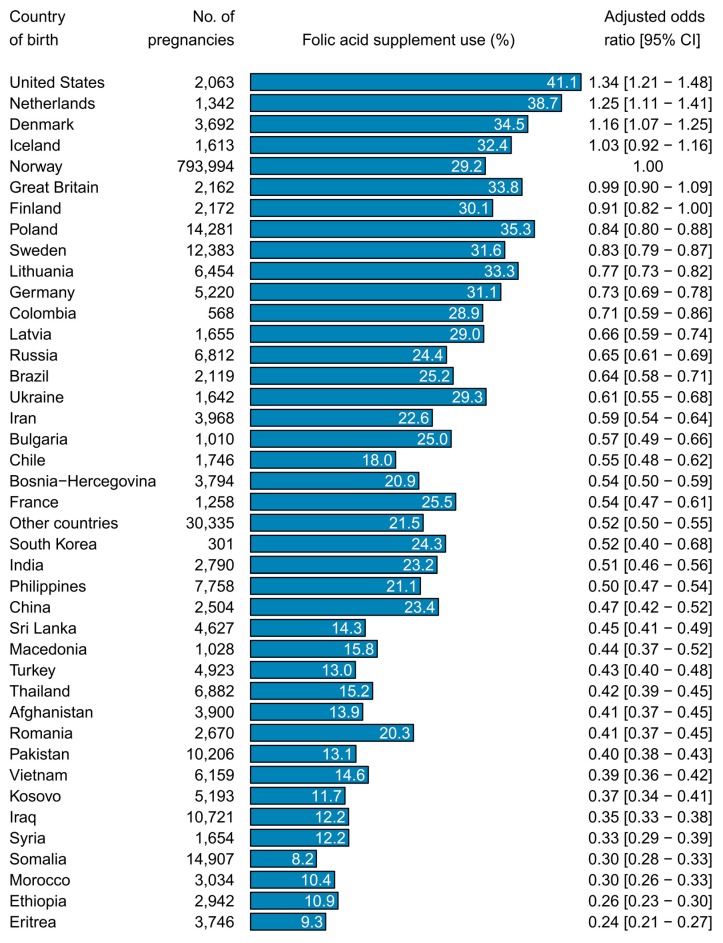
Percentages and odds ratios for preconception folic acid supplement use in 1st generation immigrant women by maternal country of birth (1999–2016). The reference group was Norwegian-born women with two Norwegian-born parents (non-immigrants; Norway). Odds ratios were adjusted for year of birth, maternal age, marital status, parity, geographical region, education, and income. CI indicates confidence interval.

**Figure 3 nutrients-11-02300-f003:**
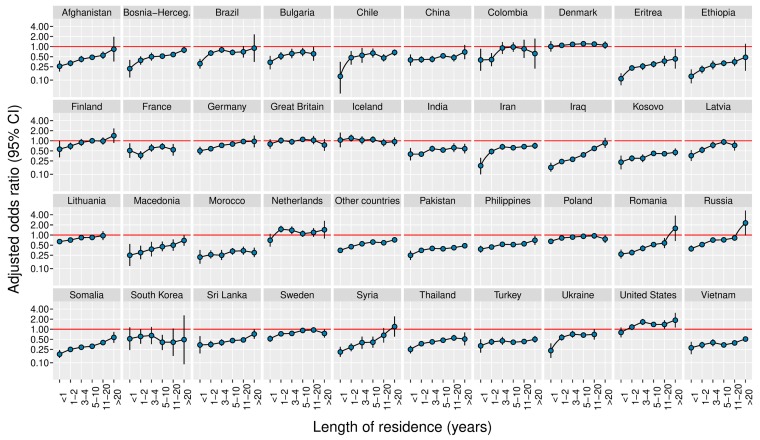
Adjusted odds ratios for preconception folic acid supplement use in 1st generation immigrant women by maternal length of residence and maternal country of birth (1999–2016). The reference group was Norwegian-born women with two Norwegian-born parents (non-immigrants). Odds ratios were adjusted for year of birth, maternal age, marital status, parity, geographical region, education, and income. CI indicates confidence interval.

**Table 1 nutrients-11-02300-t001:** Sample characteristics by maternal immigrant category (1999–2016).

Characteristic	Primary Comparison Groups			Other Groups with Mixed Background		
	Non-Immigrant Women ^a^	2nd Generation Immigrant Women ^b^	1st Generation Immigrant Women ^c^	Norwegian-born Women with One Foreign-born Parent	Foreign-born Women with Two Norwegian-born Parents	Foreign-born Women with one Norwegian-born Parent
No. of pregnancies	793,994	7965	202,234	34,576	10,211	6906
Maternal age at birth, years (mean ± SD)	29.7 ± 5.1	27.6 ± 4.8	29.9 ± 5.3	29.7 ± 5.3	30.3 ± 5.3	30.5 ± 5.5
Single status (widowed/divorced/other) (%)	58,643 (7.4%)	670 (8.4%)	17,016 (8.4%)	3267 (9.4%)	919 (9.0%)	762 (11.0%)
Educational level (%) ^d^						
No education	4 (<1%)	9 (0.1%)	4426 (3.0%)	3 (<1%)	0 (0.0%)	1 (<1%)
Primary education	130,676 (16.5%)	2324 (30.2%)	44,413 (29.9%)	6589 (19.2%)	1605 (15.8%)	1264 (18.7%)
Secondary education	266,113 (33.6%)	2615 (34.0%)	37,032 (24.9%)	10,080 (29.3%)	2868 (28.2%)	1984 (29.4%)
University/college	396,137 (50.0%)	2753 (35.7%)	62,633 (42.2%)	17,710 (51.5%)	5708 (56.1%)	3508 (51.9%)
Educational level, missing (%)	1064 (0.1%)	264 (3.3%)	53,730 (26.6%)	194 (0.6%)	30 (0.3%)	149 (2.2%)
Income level, percentiles (%) ^d,e^						
<25 percentile	168,429 (22.3%)	1833 (25.2%)	55,989 (39.0%)	7092 (21.8%)	2208 (23.0%)	1516 (23.9%)
25–50 percentile	195,891 (25.9%)	1533 (21.1%)	30,366 (21.1%)	7228 (22.2%)	2179 (22.7%)	1373 (21.6%)
50–75 percentile	197,075 (26.1%)	1766 (24.3%)	28,735 (20.0%)	8391 (25.8%)	2377 (24.8%)	1585 (25.0%)
≥75 percentile	194,934 (25.8%)	2142 (29.4%)	28,491 (19.8%)	9806 (30.2%)	2839 (29.6%)	1874 (29.5%)
Income level, missing (%)	37,665 (4.7%)	691 (8.7%)	58,653 (29.0%)	2059 (6.0%)	608 (6.0%)	558 (8.1%)
Primiparous birth (%)	328,699 (41.4%)	4047 (50.8%)	84,126 (41.6%)	15,608 (45.1%)	4425 (43.3%)	2909 (42.1%)
Health region (Norway) ^d^						
South-East	405,806 (51.1%)	7055 (88.6%)	132,822 (65.8%)	22,925 (66.4%)	6330 (62.1%)	4582 (66.4%)
West	188,685 (23.8%)	687 (8.6%)	37,754 (18.7%)	6550 (19.0%)	2230 (21.9%)	1314 (19.1%)
Middle	118,945 (15.0%)	145 (1.8%)	18,973 (9.4%)	2847 (8.2%)	1020 (10.0%)	588 (8.5%)
North	79,966 (10.1%)	72 (0.9%)	12,379 (6.1%)	2219 (6.4%)	621 (6.1%)	412 (6.0%)
Health region (Norway), missing (%)	592 (0.1%)	6 (0.1%)	306 (0.2%)	35 (0.1%)	10 (0.1%)	10 (0.1%)
Length of residence, years (mean ± SD) ^f^			6.3 ± 6.3			
Folic acid supplement use (%)	231,815 (29.2%)	2029 (25.5%)	42,792 (21.2%)	10,649 (30.8%)	2996 (29.3%)	2115 (30.6%)

SD, standard deviation. ^a^ Norwegian-born women with two Norwegian-born parents. ^b^ Norwegian-born women with two foreign-born parents. ^c^ Foreign-born women with two foreign-born parents. ^d^ Percentage of non-missing data. ^e^ Quartiles estimated for the total study period 1999-2016. ^f^ Excluded were 2,020 due to implausible data.

**Table 2 nutrients-11-02300-t002:** Percentages and odds ratios for preconception folic acid supplement use in 1st generation immigrant women according to seven Global Burden of Disease super regions (1999–2016).

Global Burden of Disease Super Regions ^a^	No. of Pregnancies	Folic Acid Supplement Use, No. (%)	Crude Odds Ratio [95% CI]	Adjusted Odds Ratio [95% CI] ^b^
Non-immigrant women ^c^	793,994	231,815 (29.2)	1.00	1.00
Central Europe, Eastern Europe, and Central Asia	51,499	14,074 (27.3)	0.91 [0.89–0.93]	0.66 [0.64–0.67]
High-income	39,145	12,483 (31.9)	1.14 [1.11–1.16]	0.86 [0.84–0.88]
Latin America and Caribbean	5678	1436 (25.3)	0.82 [0.77–0.88]	0.64 [0.60–0.68]
North Africa and Middle East	32,289	4434 (13.7)	0.39 [0.37–0.40]	0.39 [0.38–0.41]
South Asia	13,631	2091 (15.3)	0.44 [0.42–0.46]	0.43 [0.41–0.45]
Southeast Asia, East Asia, and Oceania	30,123	5246 (17.4)	0.51 [0.50–0.53]	0.45 [0.43–0.46]
Sub-Saharan Africa	29,869	3028 (10.1)	0.27 [0.26–0.28]	0.31 [0.30–0.32]

CI, confidence interval. ^a^ Seven Global Burden of Disease super regions [24]. ^b^ Adjusted for year of birth, maternal age, marital status, parity, geographical region, education, and income. ^c^ Reference group: Norwegian-born women with two Norwegian-born parents.

**Table 3 nutrients-11-02300-t003:** Percentages and odds ratios for preconception folic acid supplement use by maternal length of residence among 1st generation immigrant women (1999–2016).

Length of Residence (Years)	No. of Pregnancies ^a^	Folic Acid Supplement Use, No. (%)	Crude Odds Ratio [95% CI]	Adjusted Odds Ratio [95% CI] ^b^
Non-immigrant women ^c^	793,994	231,815 (29.2)	1.00	1.00
<1	18,781	2556 (13.6)	0.38 [0.37–0.40]	0.39 [0.37–0.41]
1–2	48,144	8880 (18.4)	0.55 [0.54–0.56]	0.50 [0.49–0.52]
3–4	36,112	8234 (22.8)	0.72 [0.70–0.73]	0.60 [0.58–0.61]
5–10	58,990	14,071 (23.9)	0.76 [0.74–0.78]	0.63 [0.62–0.65]
11–20	28,775	6370 (22.1)	0.69 [0.67–0.71]	0.62 [0.60–0.64]
>20	9412	2552 (27.1)	0.90 [0.86–0.95]	0.63 [0.60–0.67]

CI, confidence interval. ^a^ Excluded were 2,020 due to implausible data. ^b^ Adjusted for year of birth, maternal age, marital status, parity, geographical region, education, and income. ^c^ Reference group: Norwegian-born women with two Norwegian-born parents.

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
