# Peer review of "Preconception Folic Acid Supplement Use in Immigrant Women (1999–2016)"

_nutrients, 2019, doi:10.3390/nu11102300_

Round 1

Reviewer 1 Report

This study investigated preconception folic acid in immigration populations in Norway. 

Overall, I think the study is well written and the data is analyzed well.

Did the researchers conduct any follow-up with the women after birth to analyze prevalence of NTDs?

Author Response

Response to Reviewer 1 Comments

Point 1:  Did the researchers conduct any follow-up with the women after birth to analyze prevalence of NTDs?

Response 1: No, in this study, we decided to focus solely on immigrant women’s folic acid supplement use and compliance with the official recommendations. However, in the near future, we will also follow-up our sample to conduct a similar analysis of NTD prevalence in immigrant women.

Reviewer 2 Report

This paper investigates the relationship between Folic acid use with immigrant category and length of residence using a large population data from Norway. The authors have carefully considered appropriate tests for their study, performing multiple imputations and using robust errors to strengthen the analysis.

Overall the paper is well presented. There are some minor comments listed below:

The authors mentioned that there may be multiple pregnancies by the same mother, and robust standard errors were used to account for within-mother clustering.

- What is the proportion of multiple pregnancies?

- Could you elaborate more on which type of standard errors were used?

- and why was this chosen instead of Generalised linear mixed-effects model or Generalised estimating equations?

- Was there computational difficulties in fitting Logistic regression to such a large dataset?

Regarding multiple imputation:

- Why was Predictive mean matching (which may lead to biased results in certain situations) preferred over methods such as Chained equations?

- Were there other auxiliary variables used in the imputation process?

- Also, an m=5 number of imputations seems to be rather low.

It would also be interesting to test possible interaction effects between country and length of residence and explore whether the effect of length of residence differs by country/region.

Author Response

Response to Reviewer 2 Comments

Point 1: The authors mentioned that there may be multiple pregnancies by the same mother, and robust standard errors were used to account for within-mother clustering. - What is the proportion of multiple pregnancies?

Response 1: The proportion of multiple births during the period 1999-2016 was estimated to be 1.77% (19,048 multiple births / 1,075,444 all births). We have added this proportion on line 74.

Point 2: - Could you elaborate more on which type of standard errors were used?

Response 2: We used the clustered Sandwich estimator based on the Sandwich-package in R (vcovCL). We first performed full separate logistic regression analysis on each of the five completed/imputed dataset. We then ran the Sandwich estimator for each analysis before combining the estimates using Rubin’s combination rules. We have updated the description of standard error estimation on line 129 and the description of the pooling of estimates on lines 135-136.

Point 3: - and why was this chosen instead of Generalised linear mixed-effects model or Generalised estimating equations?

Response 3: In our study, we had 612,054 clusters of varying size (i.e., women with one or more pregnancies). In our experience, the GEE methodology and, particularly, the mixed-effects models require a huge amount of time and computer power to handle matrix operations with so many clusters. This is not a problem with cluster-robust standard error estimation, because the correction of standard errors is applied to the model-fitted standard errors after performing logistic regression.

Point 4: - Was there computational difficulties in fitting Logistic regression to such a large dataset?

Response 4: No, we had no difficulties in fitting logistic regression models to these data.

Point 5: Regarding multiple imputations: - Why was Predictive mean matching (which may lead to biased results in certain situations) preferred over methods such as Chained equations?

Response 5:  By using the PMM algorithm, we did not need to define models for the distributions of the missing values. As such, the PMM method is less vulnerable to model misspecification than other methods. The PMM also performs well in large datasets and has several properties that make this method as a good choice for multiple imputations (see for example conclusions in https://stefvanbuuren.name/fimd/sec-pmm.html).

Point 6: - Were there other auxiliary variables used in the imputation process?

Response 6: The imputation model included the same variables as those contained in the final analytical models. We did not include additional auxiliary variables. The final analytical models are described in the statistics section. We have updated the text on lines 134-135.

Point 7: - Also, an m=5 number of imputations seems to be rather low.

Response 7: We agree that five imputations may seem low. However, in our experience, standard error estimates seem to vary little from imputation to imputation in large datasets like ours. Below, we have compared standard errors (SE) of log odds estimates of the original five imputations with the same estimates using 10 imputations. We used covariate-adjusted estimates for the seven GBD super regions as an example (i.e., Table 2). As seen from the table, there were essentially no differences in pooled log odds or SE estimates between imputation numbers (m = 5 and m = 10).

Table R1. Pooled log odds and pooled SE with clustered Sandwich estimation.

GBD super regions

Log odds (SE)

(with 5 imputations)

Log odds (SE)

(with10 imputations)

Central Europe, Eastern Europe,

and Central Asia

-0.416 (0.011)

-0.416 (0.011)

High-income

-0.152 (0.012)

-0.152 (0.012)

Latin America and Caribbean

-0.446 (0.032)

-0.446 (0.032)

North Africa and Middle East

-0.931 (0.018)

-0.932 (0.017)

South Asia

-0.851 (0.025)

-0.851 (0.025)

Southeast Asia, East Asia, and Oceania

-0.804 (0.016)

-0.804 (0.016)

Sub-Saharan Africa

-1.177 (0.021)

-1.178 (0.021)

Point 8: It would also be interesting to test possible interaction effects between country and length of residence and explore whether the effect of length of residence differs by country/region.

Response 8: Although we did not perform a formal statistical test of the country-by-length interaction, we believe the proposed analyses are already conducted in Figure 3.